



# An "island" in the stratosphere – On the enhanced annual variation of water vapour in the middle and upper stratosphere in the southern tropics and subtropics

Stefan Lossow[1], Hella Garny[2], and Patrick Jöckel[2]

[1]Karlsruhe Institute of Technology, Institute for Meteorology and Climate Research, Hermann-von-Helmholtz-Platz 1, 76344 Leopoldshafen, Germany.
[2]Deutsches Zentrum für Luft- und Raumfahrt (DLR), Institut für Physik der Atmosphäre, 82234 Oberpfaffenhofen-Wessling, Germany.

*Correspondence to:* Stefan Lossow (stefan.lossow@kit.edu)

**Version: Sunday 7th May, 2017 21:02 CET**

**Abstract.** The annual variation of water vapour exhibits a distinct isolated maximum in the middle and upper stratosphere in the southern tropics and subtropics, peaking typically around 15°S in latitude and close to 3 hPa ($\sim$40.5 km) in altitude.

This enhanced annual variation is primarily related to the Brewer-Dobson circulation and hence also visible in other trace gases. So far this feature has not gained much attention in the literature and the present work aims to add more prominence. Using Envisat/MIPAS (Environmental Satellite / Michelson Interferometer for Passive Atmospheric Sounding) observations and ECHAM/MESSy (European Centre for Medium-Range Weather Forecasts Hamburg/Modular Earth Submodel System) Atmospheric Chemistry (EMAC) simulations we provide a dedicated illustration and a full account for the reasons of this

enhanced annual variation.

## 1 Introduction

Water vapour is one of the most important trace constituents in the stratosphere. It owes this importance to its role as a greenhouse gas, its impact on the stratospheric ozone budget and its usefulness as a tracer of dynamical processes. As the most important greenhouse gas in the lower stratosphere, water vapour plays a decisive role for the global warming at the

Earth's surface (Riese et al., 2012). In that context, water vapour is part of an important feedback mechanism. A warmer surface climate leads to an increase in stratospheric water vapour, which causes the surface temperatures to increase further. Dessler et al. (2013) estimated this water vapour feedback to be about $0.3\,\mathrm{W\cdot m^{-2}}$ for a $1\,\mathrm{K}$ temperature anomaly in the middle troposphere. For the ozone budget, water vapour is important in different ways. On one hand, it is an essential component of polar stratospheric clouds (PSCs). The heterogenous chemistry taking place on the cloud particle surfaces is responsible for

the severe ozone depletion in the polar lower stratosphere during winter and spring, in particular in the Antarctic. On the other hand, water vapour is also the primary source of hydrogen radicals ($HO_x = OH$, H, $HO_2$) in the stratosphere. These radicals destroy ozone within auto-catalytic cycles, being especially important in the lower and the uppermost part of the stratosphere





(Brasseur and Solomon, 2005). Since the chemical life time of water vapour is in a similar order to the typical time scales of dynamical processes, it can be used to diagnose such processes. In the stratosphere this concerns of course primarily the Brewer-Dobson circulation as highlighted by the tape recorder in the tropical lower stratosphere (Mote et al., 1996).

The water vapour budget in the stratosphere is determined by a number of processes. The input from the troposphere, as well as the in-situ oxidation of methane and molecular hydrogen, are the major sources of stratospheric water vapour. The reaction of water vapour with $O(^1D)$ and photodissociation are most important sink processes. In general the water vapour volume mixing ratio increases with altitude in the stratosphere, primarily due to the dominant role of the methane oxidation (Le Texier et al., 1988). The efficiency of this processes increases with altitude and maximises typically in the upper stratosphere. Around the stratopause the water vapour volume mixing ratio generally exhibits a pronounced maximum in the vertical distribution, indicating an equilibrium between all source and sink processes.

Due to the importance of water vapour in the stratosphere, the research has a large focus on understanding the long-term evolution and the drivers behind it. This concerns changes of the input from the troposphere, abundances of methane, dynamical processes as well as the influence of the Sun or volcanic eruptions (e.g. Oltmans et al., 2000; Rosenlof et al., 2001; Randel et al., 2006; Hurst et al., 2011; Hegglin et al., 2014; Dessler et al., 2014; Urban et al., 2014; Eichinger et al., 2015; Schieferdecker et al., 2015; Tao et al., 2015; Brinkop et al., 2016; Löffler et al., 2016; Dessler et al., 2016). In the present work we focus on variability on a much shorter time scale, namely the annual variation. In the middle and upper stratosphere in the southern tropics and subtropics a pronounced and isolated maximum in the amplitude of the annual variation can be observed (which we for simplicity denote as enhanced annual variation). This region is normally associated with the semi-annual variation that prominently maximises in the tropical upper stratosphere (Carr et al., 1995; Jackson et al., 1998). The enhanced annual variation as such has been observed in the past and some references exist in the literature. Most prominently Jackson et al. (1998) showed this in observations from the HALOE (Halogen Occultation Experiment) instrument aboard UARS (Upper Atmosphere Research Satellite). Other instances can be found for example in the work of Holton and Choi (1988) or Randel et al. (1998), which show this feature in methane. Since $H_2O + 2 \cdot CH_4$ is approximately a conserved parameter in the stratosphere and lower mesosphere (Le Texier et al., 1988; Siskind and Summers, 1998) the structure and variability of methane and water vapour are tightly linked with each other. All these publications have, however, a much wider scientific focus: variability in general and atmospheric transport. This motivated us to specifically highlight the enhanced annual variation in the southern tropics and subtropics. On one hand, we aim to provide a dedicated description of this feature. On the other hand, we aim to provide a full attribution of the reasons for this feature, that has been associated to the Brewer-Dobson circulation (Jackson et al., 1998). To do so we utilise both satellite observations and model simulations. The former we use primarily for the description and characterisation, while the latter are employed for the explanation and attribution of the feature. The observational results are mostly based on Envisat/MIPAS nominal mode measurements (Fischer et al., 2008). Those were performed almost daily, typically covering the altitude range from the upper troposphere to the lower mesosphere. We focus on the time period from January 2005 to the end of the MIPAS mission in April 2012 during which the observations were based on the same spectral resolution (Fischer et al., 2008). The data were retrieved with the IMK/IAA processor, which is a collaboration between the "Institut für Meteorologie und Klimaforschung" (IMK) in Karlsruhe/Germany and the "Instituto de Astrofísica de Andalucía"





(IAA) in Granada/Spain (von Clarmann et al., 2009). On the simulation side, we use on results from the EMAC model (Jöckel et al., 2016). Those are based on the REF-C1-SD (transient hindcast reference simulation with specified dynamics) scenario as defined by the SPARC (Stratosphere-troposphere Processes And their Role in Climate) Chemistry-Climate Model Initiative (Eyring et al., 2013). The specified dynamics data is taken from the Interim ECMWF (European Centre for Medium-Range Weather Forecasts) Reanalysis project (ERA-interim, Dee et al., 2011), however water vapour itself was not nudged (Jöckel et al., 2016). The simulations employed here cover the the time period from 2008 to 2012. They are equivalent to those labeled "RC1SD" in the work of Brinkop et al. (2016) with two exceptions. On one hand, the use of specified dynamics is extended up to 1 hPa (instead of 10 hPa). On the other hand, the simulations here contain additional diagnostic tracers for the chemical production and loss of water vapour ("ProdH2O" and "LossH2O"). Those tracers are accumulated in every chemical reaction if water vapour is produced or destroyed. Analogous tracers for ozone are documented by Jöckel et al. (2016). All observational and model results and analyses in this work are based on monthly and zonal mean data.

In the next section we describe and characterise the enhanced annual variation in the middle and upper stratosphere in the southern tropics and subtropics based on the satellite observations. A detailed discussion of the reasons for this feature is subsequently provided in Sect. 3.

## 2 Description

Figure 1 shows the amplitude (left panel) and the phase (right panel) of the annual variation in water vapour as function of latitude and pressure derived from the MIPAS IMK/IAA v220/221 data set (Schieferdecker et al., 2015). The right axes of the panels indicate the log-pressure height using an atmospheric scale height of 7 km. All geometric altitudes noted in this manuscript are based on this quantity. The characteristics of the annual variation have been obtained by means of regression, employing a regression model containing an offset, a linear term, the semi-annual and annual variation as well a QBO term (for more details please see Lossow et al., 2017). The largest stratospheric annual variability is found in the polar regions and close to the tropopause. In the polar regions this variability is primarily due to the strong annual variation of the vertical advection. In the Antarctic lower stratosphere the annual variation of dehydration contributes significantly to the annual variation of water vapour. The large variability close to the tropopause is caused by variations of the tropopause height and the finite vertical resolution of the data retrieved from the satellite observations (which is typically between 3 km and 4 km). The only other region exhibiting a substantial annual variation is located in the middle and upper stratosphere in the tropics and subtropics. Due to its isolated nature, this feature resembles an island in the ocean, provoking the term used in the manuscript title. The annual variation maximises at about 15°S in the altitude range range between 3 hPa and 2 hPa (~40.5 km – 43.5 km). The MIPAS data exhibit an overall peak amplitude of 0.58 ppmv. The data sets compared within the second SPARC water vapour assessment (WAVAS-II) indicate on average a peak amplitude of almost 0.65 ppmv and a corresponding standard deviation of 0.1 ppmv (not shown here, Lossow et al., 2017). The white contour line in Fig. 1 indicates where the amplitude of the enhanced annual variation has decreased to the half of its peak amplitude. Based on this criterion the enhanced annual variation extends from 38°S to 2°N in latitude and from 6 hPa to 1 hPa (~38 km – 48.5 km) in altitude. The phase of the enhanced annual



variation shows a distinct latitude dependence, with the annual maximum shifting from July close to the Equator to September in the subtropics.

There is a corresponding feature in the Northern Hemisphere, however distinctively weaker and more towards the north. The MIPAS observations indicate a peak variation of somewhat more than 0.2 ppmv between 20°N to 25°N slightly below 4 hPa

(∼38.5 km). Different data sets exhibit a distinct variation in the peak location, in particular with respect to the altitude. Also, the location is rather dependent on the actual approach chosen to derive the annual variation (see Sect. 5.3 of Lossow et al., 2017).

To illustrate the special character of the enhanced annual variation in water vapour in the middle and upper stratosphere, Fig. 2 shows the characteristics of the annual variation in the mesosphere. These results are based on observations of the

Sub-Millimetre Radiometer (SMR) aboard the Odin satellite (Murtagh et al., 2002) from 2001 to 2014. More information on the SMR water vapour measurements can be found in Lossow et al. (2008). Below about 0.005 hPa (∼85.5 km) the largest annual variation is found in the polar regions, very similar to the stratosphere. However, no isolated maximum can be found in the southern tropics and subtropics. Instead, the annual variation typically decreases towards lower latitudes and actually minimises around 15°S at altitudes above 0.12 hPa (∼63 km). This indicates a clear inter-hemispheric asymmetry primarily

induced by differences in the forcing of the mesospheric pole-to-pole circulation.

In Fig. 3 the MIPAS water vapour time series for the latitude band between 18.5°S and 13.5°S is shown. In this latitude band the MIPAS observations show the overall peak amplitude of the enhanced annual variation, namely at pressure level of about 2.7 hPa (∼41.5 km). This level is indicated by the red dashed line in the figure. The observations show that at altitudes around 2.7 hPa (∼41.5 km) the annual minimum typically occurs in the first months of the year (austral summer/autumn). The annual

maxima, on the other hand, can be in general observed early in the second half of the year (austral winter/spring), as shown in the right panel of Fig. 1. At 2.7 hPa (∼41.5 km) also some apparent semi-annual variation is visible, in particular in 2007 and 2010. In 2007 the annual maximum occurs a little bit later than normal, while in 2010 it appears already late in the first half of the year. Higher up, between 1.5 hPa and 1 hPa (∼45.5 km – 48.5 km) the semi-annual variation is even more pronounced.

For a final description, Fig. 4 shows the latitudinal cross section of the water vapour time series at a pressure level of

2.7 hPa (∼41.5 km). A red dashed line is drawn at 16°S to indicate the latitude where the enhanced annual variation exhibits its peak amplitude in the Southern Hemisphere. For the sake of comparison, a dark blue dotted line is drawn at the corresponding latitude in the Northern Hemisphere. The figure indicates a clear inter-hemispheric asymmetry in the tropical band. The extreme volume mixing ratios have a clear tendency to be shifted towards the Southern Hemisphere. The low volume mixing ratios at 15°S during austral summer and autumn extend towards the Antarctic where those can be observed in winter. The same kind of

behaviour can also be observed for the high volume mixing ratios just with the corresponding time shift. Such an overall pattern is however not as obvious in the Northern Hemisphere, illustrating once more inter-hemispheric differences in the processes giving rise to the water vapour distribution at this altitude. Beyond that, a strong inter-annual variability is visible in the tropical water vapour volume mixing ratios. Most prominently the annual minimum exhibits very low ratios in 2006, 2008 and 2011, while in the 2007 and 2009 the ratios are clearly higher. This variability is predominantly related to the QBO that has a local maximum in this region (Randel et al., 1998; Lossow et al., 2017).



## 3 Discussion

The analysis of the annual variation amplitude in water vapour shows an isolated maximum in the southern tropics and sub-
tropics in the middle and upper stratosphere. The annual variation typically peaks at about 15°S in latitude and between 3 hPa
and 2 hPa in altitude in the observations. In the following discussion we focus first on the attribution of this enhanced annual
variation. Initially this will be based on additional observational results. For a more advanced attribution we use the model
results from EMAC. In the Northern Hemisphere a weaker counterpart of the enhanced annual variation in the south can be ob-
served. This counterpart typically peaks at higher latitudes and lower altitudes than in the Southern Hemisphere. In the second
part of the discussion will focus on the reasons for this inter-hemispheric asymmetry.

### 3.1 Attribution

In principle both chemical and dynamical reasons or their combination could explain this enhanced annual variation. To start
the attribution discussion of this feature, Fig. 5 shows the amplitude of the annual variation for other longer-lived trace gases
observed by MIPAS (Plieninger et al., 2016; Eckert et al., 2016; Chirkov et al., 2016), again as function of latitude and altitude.
The results were obtained by the same approach as used for the water vapour results presented in Fig. 1. In all trace gases a
similar enhanced annual variation as seen in water vapour can be observed in the middle and upper stratosphere in the southern
tropics and subtropics. The existence in methane could be expected due to its tight chemical relationship to water vapour, in
line with previous observations reported by Holton and Choi (1988) and Randel et al. (1998). The existence of the feature in
the other trace gases clearly indicates that dynamics plays a larger role than methane chemistry itself for the explanation of the
enhanced annual variation in water vapour. The exact location of the peak variation differs a bit for the different trace gases,
implying an importance of the actual trace gas gradients. For both, nitrous oxide ($N_2O$) and CFC-12 ($CCl_2F_2$), the annual
variation peaks at 12°S between 5 hPa and 6 hPa (∼36 km – 37 km). For HCFC-22 ($CHClF_2$) the peak is located at 18°S and
close to 3 hPa (∼40.5 km). This location is probably affected by the coarse vertical resolution of the HCFC-22 data, close to
the upper altitude limit where a reasonable retrieval is possible. The resolution exceeds 10 km altitudes (see Fig. 2 and 3 of
Chirkov et al., 2016), explaining why the enhanced annual variation is much more extended in the altitude domain than for the
other trace gases.

Figure 6 shows the MIPAS time series of water vapour (blue, left axis) for the latitude band between 18.5°S and 13.5°S
and 2.7 hPa (∼41.5 km) in relation to the corresponding time series of the other longer-lived trace gases considered in the last
figure. In all panels of the figure a pronounced anti-correlation between water vapour and the other trace gases can be observed.
The correlation coefficients are between -0.87 and -0.96 for the comparisons with methane, nitrous oxide and CFC-12. The
comparison with the HCFC-22 time series yields a lower correlation coefficient, i.e. -0.59. As argued above this reduction is
probably due to the coarse vertical resolution of the HCFC-22 data, while the resolution of the water vapour data is about
5 km. While the water vapour volume mixing ratio is increasing with altitude in the middle and upper stratosphere, the ratio
of the other trace gases is decreasing with altitude. Thus, negative correlation coefficients in the comparisons clearly hint





towards vertical advection as one of the main reasons for the enhanced annual variation of water vapour, in agreement with the considerations by Jackson et al. (1998).

From the observations presented in the two previous figures it has been possible to discern the important role of vertical advection for the enhanced annual variation in water vapour. In the further discussion we use EMAC simulation results, as described in the Introduction. For comparison with the observations (see Fig. 1) Fig. 7 shows the amplitude and phase of the annual variation as derived from the simulations. Overall, a good agreement between the observations and simulations is found. In the simulations, the enhanced annual variation also extends from 38°S to 2°N in latitude. In altitude, the feature ranges from 10 hPa to 1.5 hPa (∼32.5 km – 45.5 km), indicating a larger vertical extent and a slight downward shift compared to the observations. The simulations exhibit the peak variation at 15°S and 3.2 hPa (∼40 km). The peak amplitude is 0.51 ppmv. This is somewhat lower than in the MIPAS observations, but still within the range of the different observational data sets compared in the second SPARC water vapour assessment (Lossow et al., 2017). The amplitude agreement at 15°S is within ± 0.05 ppmv right below the peak altitude down to about 6 hPa. Above and below this altitude region the amplitude differences between the observations and simulations are larger, consistent with the differences in the vertical extent of the enhanced annual variation. In terms of the phase the observations and simulations agree within ±1 month. However, there is a tendency that the annual maximum occurs later in the simulations. For the northern hemispheric counterpart of the feature, similar differences in the amplitude and phase can be found. Interestingly, the annual maximum derived from simulations has here a tendency to occur earlier than in the observations. Overall, this is arguably not a perfect comparison. Neither in terms of the temporal and spatial consistency between the observations and simulations, nor in terms of the quantitative agreement. Nonetheless, the generally good agreement between the simulations and observations allows the attribution of the enhanced annual variation with a certain degree of confidence from the simulation side.

Figure 8 shows the EMAC water vapour time series at 15°S. In the upper panel contours of the residual vertical velocity are overlaid, in the lower panel the residual meridional velocity is considered. Positive winds are given by red contours, negative winds by white contours. Overall, the water vapour structure in EMAC is very similar to that in the observation (see Fig. 3). However, there are differences in the absolute amount of water vapour, with the simulations showing lower volume mixing ratios than the observations. As concluded from Fig. 5 and 6, some coherent behaviour between the temporal structure of water vapour and the vertical velocity can be found. Increases/decreases in water vapour roughly go along with upwelling/downwelling. At the peak altitude of the enhanced annual variation (i.e. 3.2 hPa) the correlation coefficient between water vapour and the vertical velocity amounts to –0.49. A more detailed look indicates that increases of water vapour actually precede downwelling by approximately 2 months at the peak altitude. Higher up, at 1 hPa, no such delay is observed. Instead the delay between increases of water vapour and downwelling increases with decreasing altitude. This clearly indicates that other processes play also a role for the annual variation besides the vertical advection. Decreases of water vapour are more directly related to upwelling. Occasionally there is also a delay between these two quantities, in particular below the peak altitude, but not as obvious as between water vapour increases and downwelling. Beyond that some apparent QBO variation can be found in the downwelling, extending to lower altitudes in 2008, 2010 and 2012 as compared to 2009 and 2011.





Also between the residual meridional velocity and the water vapour distribution some coherent behaviour can be observed. Most prominently, the decrease in water vapour from the annual maximum to the minimum is typically accompanied by northward winds. Southward winds prevail during the time of the annual maximum. In addition, increases of water vapour below 2 hPa ($\sim$43.5 km) coexist with southward winds. At the peak altitude of the enhanced annual variation, the correlation between the residual meridional wind and water vapour is not statistically significant (correlation coefficient is –0.02). In summary, it can be concluded that also the meridional advection contributes to the enhanced annual variation feature. However this does not universally apply throughout the year. The meridional advection influence is greatest during austral summer.

For a more detailed attribution Fig. 9 shows the EMAC water vapour budget at the peak location of the enhanced annual variation at 15°S and 3.2 hPa, using the tendencies of the continuity equation given below:

$$
\underbrace{\frac{\partial \overline{\chi}}{\partial t}}_{\text{total tendency}} = \underbrace{-\frac{\overline{v}^*}{a} \cdot \frac{\partial \overline{\chi}}{\partial \phi}}_{\text{meridional advection tendency}}
$$
$$
+ \underbrace{-\overline{w}^* \cdot \frac{\partial \overline{\chi}}{\partial z}}_{\text{vertical advection tendency}}
$$
$$
+ \underbrace{P - L}_{\text{production and loss tendency}}
$$
$$
+ \underbrace{R}_{\text{residual tendency}} . \tag{1}
$$

Here $\overline{\chi}$ denotes the water vapour zonal mean volume mixing ratio, $t$ is the time, $\phi$ the latitude and $z$ the altitude. $a =$ 111195 m·deg$^{-1}$ gives the conversion between a distance in latitude and the geometric scale. $\overline{v}^*$ and $\overline{w}^*$ describe the residual meridional and vertical velocity, respectively. $P - L$ represents the chemical production minus loss rate. The residual tendency $R$ is the tendency due to eddy transport (i.e. mixing, see Andrews et al., 1987 their Eq. 9.4.13). This tendency is not calculated explicitly here, instead we calculate the residual from the continuity equation. Consequently, a small contribution arising from uncertainties in the budget reconstruction may be included in the residual tendency. The data shown in Fig. 9 are based on averages over the time period from 2008 to 2012. The temporal behaviour of the total tendency is best resembled by that of the vertical advection tendency. The most prominent differences concern the overall amplitude of the annual variation and the time when the annual extrema occur. For the vertical advection tendency the amplitude is larger and its annual extrema occur 1 month later than for the total tendency. This confirms the dominant role of vertical advection for the enhanced annual variation of water vapour in this region, but it also indicates that the other tendencies have some importance for the exact characteristics of this feature. In particular, in austral summer the other tendencies exhibit positive values, which both weakens the overall annual variation of water vapour and adjusts the time of the annual minimum in the total tendency. During austral winter the residual tendency is negative, contributing also distinctively to the reduction of the annual variation. The time of annual maximum in the total tendency is adjusted again by a combination of meridional advection, chemistry and eddy transport.

Figure 10 provides an overview of the vertical structure of the different tendencies at 15°S. The total tendency does not exhibit much altitude variation in its sign. Exceptions occur from July to November, resulting in a prominent semi-annual





variation above about 2 hPa (∼43.5 km), in agreement with the observations (see Fig. 3). The vertical advection tendency is also rather constant in sign for the considered altitude range. While the largest positive tendencies occur close to the peak altitude of the enhanced annual variation, the largest negative tendencies can be observed a bit higher up. The temporal shift of the annual minimum to its equivalent in the total tendency is visible over a larger altitude range. For the annual maximum

the corresponding shift is altitude-dependent and minimises around 1 hPa (∼48.5 km). This is in agreement with the discussion of Fig. 8 regarding the delay between water vapour increases and downwelling. The meridional advection tendency shows a prominent altitude variation of its sign from March to June. In December and January the tendencies are positive at most altitudes, contributing to the reduction of the annual variation. The tendency of the chemical production and loss is positive at all altitudes. The largest tendencies can be observed around 2 hPa (∼43.5 km) during the first months of the year. The residual

tendency is primarily negative above 1 hPa (∼48.5 km) and primarily positive below 5 hPa (∼37 km). In-between the tendencies are negative from May to August and positive for the rest of the year. In summary, there is some altitude dependence in the budget of the enhanced annual variation at 15°S. While the vertical advection clearly remains the dominant process, especially the relative importance of the meridional advection and the eddy transport varies with altitude. However, the reduction of the annual variation in austral summer combined by meridional advection, eddy transport and chemistry continues to be a general

characteristic.

Based on the discussion so far it is clear that advection, chemistry and eddy transport all contribute to the enhanced annual variation. In the next step we assess which combination of processes is essential to reproduce the enhanced annual variation and its structure and which processes just cause minor adjustments to the overall picture. For that we compare the annual variation of the total tendency with the variation of the individual tendencies and their combinations. The results are shown in

Fig. 11. As expected, the annual variation of the total tendency exhibits the same enhanced annual variation feature as water vapour itself (panel a). The annual variation of the vertical advection tendency shows some resemblance to the total tendency (panel b). However, the amplitudes in the middle and upper stratosphere in the southern tropics and subtropics are considerably larger for the most part. These large amplitudes also clearly extend into the mid-latitudes. A similar picture can be observed in the Northern Hemisphere. Considering the annual variation of the vertical and meridional advection tendency combined

(denoted as total advection tendency) modifies the picture obtained for the vertical advection tendency only slightly (panel c). It becomes apparent that advection alone cannot explain the enhanced annual variation and its structure. Also in combination with the chemical production and loss a complete explanation is not possible (panel d). Inevitably this hints to an important role for the eddy transport described by the residual tendency. Panel (e) in Fig. 11 shows first the annual variation of this tendency individually. In the middle and upper stratosphere in the southern tropics and subtropics the variation is small and

clearly different than those seen for the other tendencies. Yet, the annual variation of the advection tendencies combined with the residual tendency yields almost the identical structure as observed for total tendency (panel f). Thus, advection and eddy transport are the essential processes to explain the enhanced annual variation and its structure, while the chemical production and loss just causes a second order modification.





## 3.2 Inter-hemispheric differences

The remaining part of the discussion will focus on the reasons for the inter-hemispheric differences. Since the vertical advection plays a major role for the enhanced annual variation a relation to the inter-hemispheric differences in the Brewer-Dobson circulation, which is weaker in austral winter, is obvious. For illustration, Fig. 12 shows the EMAC water vapour time series at 3.2 hPa (∼40 km) with contours of the residual vertical velocity overlaid in the upper panel and the residual meridional velocity in the lower panel. The red dotted line indicates again the latitude where the enhanced annual variation maximises, while the blue dotted line indicates the corresponding latitude in the Northern Hemisphere for comparison. As expected, a distinct inter-hemispheric asymmetry in the distribution of upwelling and downwelling can be observed. There are two key aspects. The weaker circulation in austral winter results in slower upwelling in the Northern Hemisphere and correspondingly in a smaller decrease of water vapour. The stronger circulation in boreal winter results in stronger upwelling in the Southern Hemisphere with a more pronounced reduction of water vapour. This explains already quantitatively a larger annual cycle in the Southern Hemisphere, as noted by Jackson et al. (1998). In addition, the weaker circulation in austral winter results in a transition from upwelling to downwelling (also denoted as the "turnaround latitude", see Rosenlof, 1995) already a bit north of the Equator, allowing the advection of moister air from above in the southern tropics and subtropics. In boreal winter, when the circulation is stronger, this transition occurs first at about 15°N. This aspect amplifies inter-hemispheric differences in the annual variation even further. It explains why the feature occurs more polewards in the Northern Hemisphere and designates the northern boundary of the enhanced annual variation.

Fig. 13 expands the discussion above a bit by showing all components of the water vapour budget as function of latitude at 3.2 hPa (∼40 km). The total tendencies at 15° latitude highlight once more that the inter-hemispheric differences concern both the annual minimum and maximum (panel a). The same picture can be also seen in the vertical advection tendency (panel b). In general the largest absolute tendencies occur in the Southern Hemisphere. In the latitude band between 5°N and 15°N the vertical advection tendency is negative almost exclusively, hampering the development of a substantial annual variation as underlined before. Also in terms of the meridional advection tendency, clear inter-hemispheric differences can be observed (panel c). At 15°S the meridional advection tendency is primarily positive while at 15°N it is more negative. In the south this tendency contributes to reduce the annual variation, while in the north it plays more a role for the semi-annual variation observable in the total tendency. There are some smaller inter-hemispheric differences in the production and loss tendency. However, this term seems to reflect primarily the solar insolation contributing to the methane oxidation as primary source of water vapour in the upper stratosphere. The residual tendency exhibits inter-hemispheric differences as well. While they are not as essential as those in the vertical advection, they evidently rank in the second place in general.

At last we address inter-hemispheric differences not at the same altitude, but at locations where the annual variation is substantial. For that Fig. 14 shows a comparison of the water vapour budget at 15°S/3.2 hPa and 24°N/5.6 hPa. The former is the location where the enhanced annual variation peaks in the EMAC simulations. The northern counterpart actually peaks at 35°N and 8.7 hPa (see Fig. 7). But with regard to the observations, which show the peak at 23°N (see Fig. 1), a lower latitude that is also representative, is considered. For a better visual comparison, the results for the Northern Hemisphere have





been shifted by 6 months. For the same reason panel (a) considers the water vapour anomaly relative to the annual mean, since the absolute volume mixing inevitably differ due to the different altitudes considered. As visible in the previous figure, there is an apparent semi-annual variation in the north, resulting in inter-hemispheric differences in the occurrence of the annual extrema. In the Southern Hemisphere substantial positive total tendencies can be observed over a longer period of time, while the level of negative tendencies is smaller in the Northern Hemisphere (panel b). In terms of the vertical advection inter-hemispheric differences are primarily visible in terms of the negative tendencies (panel c). The temporal behaviour is very much aligned. While the meridional advection tendency exhibits a lot of variation throughout the year in the Southern Hemisphere the corresponding term in the Northern Hemisphere shows a well defined annual variation (panel d). This variation is almost in anti-phase to the vertical advection tendency (shift is 1 month) leading to some substantial compensation between these terms in the Northern Hemisphere. The differences in the production an loss tendency reflect mostly the differences in the altitude considered, with the methane oxidation being more efficient at the higher altitude (panel e, see also Fig. 10). This naturally causes some differences in the relative importance of the processes giving rise to the annual variation in the two hemispheres. Also in terms of the residual tendency clear inter-hemispheric differences are visible in particular during the winter season (panel f). While in the Southern Hemisphere the eddy transport contributes to weaken the annual cycle it does the opposite in the north. At higher northern latitudes the behaviour of the residual tendency becomes more similar to the Southern Hemisphere. Overall, inter-hemispheric differences can be found in all processes giving rise to the annual variation. In addition, there are clear inter-hemispheric differences in the relative importance of the processes, with the vertical advection being less important for the total budget in the Northern Hemisphere than in the south.

## 4 Conclusions

Water vapour (and other longer lived trace gases) exhibits an isolated maximum in its annual variation amplitude in the middle and upper stratosphere in the southern tropics and subtropics. The peak variation in water vapour is typically observed around $15°$S and close to $3$ hPa ($\sim 40.5$ km). Vertical advection basically explains the temporal variation, but also meridional advection and eddy transports are needed to explain the feature and its structure. Chemistry contributes as well, but is of secondary importance. It just causes smaller adjustments, as for the example the exact amplitude of the variation. There is also a northern hemispheric counterpart of this feature, although weaker in amplitude and more shifted towards higher latitudes and lower altitudes. Hemispheric differences in the vertical transport of the Brewer-Dobson circulation, which is weaker in the Southern than the Northern Hemisphere, contribute to this inter-hemispheric asymmetry in the annual variation of water vapour. As the ascent occurs in the opposite hemisphere there is stronger upwelling in boreal winter in the Southern Hemisphere, which leads to lower water vapour volume mixing ratios than in the Northern Hemisphere in austral winter. The weaker circulation towards the south in austral winter leads to a transition from upwelling to downwelling in the middle and upper stratosphere already around the Equator. For the stronger circulation towards the north this transition occurs first around $15°$N. This explains larger volume mixing ratios during winter in the Southern Hemisphere compared to the north. In addition, it explains also the different latitude band in which the northern counterpart occurs. Also differences in other processes contribute to the inter-hemispheric





differences. The relative importance of the individual process for these differences clearly varies in time and space, occasionally outweighing the vertical advection. As a summary, Fig. 15 sketches the main aspects of the enhanced annual variation and the inter-hemispheric differences to its northern hemispheric counterpart with respect to the vertical advection within the Brewer-Dobson circulation.

The enhanced annual variation and its northern hemispheric counterpart inevitably present themselves as a natural evaluation parameter for model simulations. As the tape recorder signal in water vapour is used to assess the quality of simulations

with respect to the ascent of the Brewer-Dobson circulation in the tropical lower stratosphere, the enhanced annual variation can serve as a benchmark for the quality of simulations with regard to this circulation in the middle and upper stratosphere. Given the occurrence of this feature in multiple trace gases, with slightly different characteristics based on different gradients and chemistry, the assessment will be more rigorous than for other evaluation parameters. We have looked into the results from a number of model simulations, both using specified dynamics and free-running (not shown here). They exhibit distinct

differences in the characteristics of the enhanced annual variation and its northern hemispheric counterpart, clearly warranting further analyses. For EMAC the application of specified dynamics up to 1 hPa (see Introduction) yielded a much better agreement with the observations than the standard application up to only 10 hPa. This might highlight the need of specified dynamics up to the top of the stratosphere or even highlight the quality of the ERA-interim reanalysis data in this altitude region.

*Data availability.* The data sets used in this work can be accessed as follows:

• The MIPAS data are available on the following website: https://www.imk-asf.kit.edu/english/308.php[1].

• The SMR data can be accessed on the following website: http://amazonite.rss.chalmers.se:8280/OdinSMR/searchl2[1].

• The data of the EMAC simulations described above will be made available in the Climate and Environmental Retrieval and Archive (CERA) database at the German Climate Computing Centre (DKRZ, website: http://cera-www.dkrz.de/WDCC/ui/Index.jsp). The corresponding digital object identifiers (DOI) will be published on the MESSy consortium website (http://www.messy-interface.org).

Alternatively, the data can be obtained on request from Patrick Jöckel (Patrick.Joeckel@dlr.de).

*Competing interests.* The authors declare that they have no conflict of interest.

*Acknowledgements.* S. Lossow was funded by the DFG Research Unit "Stratospheric Change and its Role for Climate Prediction" (SHARP) under contract STI 210/9-2. H. Garny was funded by the Helmholtz Association under grant VH-NG-1014 (Helmholtz Hochschul-Nach-wuchsforschergruppe MACCClim). We would like to thank the European Space Agency (ESA) for making the MIPAS level-1b data set avail-

able. Odin is a Swedish-led satellite project funded jointly by the Swedish National Space Board (SNSB), the Canadian Space Agency (CSA), the National Technology Agency of Finland (Tekes) and the Centre National d'Etudes Spatiales (CNES) in France. The Swedish Space Cor-

---

[1]registration needed



poration has been the industrial prime constructor. Since April 2007 Odin is a third-party mission of ESA. The EMAC simulations have been performed at the German Climate Computing Centre (DKRZ) through support from the Bundesministerium für Bildung und Forschung (BMBF). DKRZ and its scientific steering committee are gratefully acknowledged for providing the high performance computing (HPC) and data archiving resources for ESCiMo (Earth System Chemistry integrated Modelling) consortial project. We thank Gerald E. Nedoluha from

the Naval Research Laboratory for valuable comments.

We acknowledge support by Deutsche Forschungsgemeinschaft and Open Access Publishing Fund of Karlsruhe Institute of Technology.





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



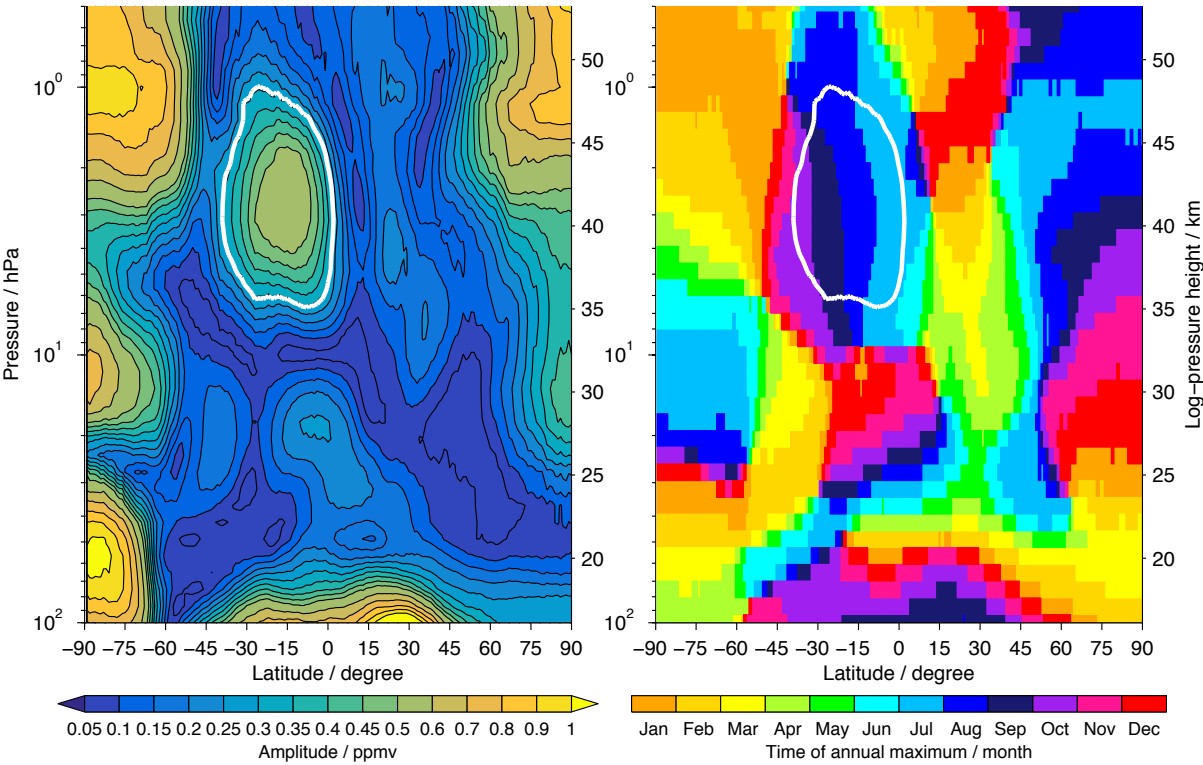

**Figure 1.** Characteristics of the annual variation in water vapour based on MIPAS observations from 2005 to 2012. The left panel shows the amplitude as function of latitude and altitude. In the right panel the phase is given, expressed by the month of a calendar year in which the annual maximum occurs. The white contour line indicates the region where the amplitude of the enhanced annual variation is half of its peak amplitude. The data set has been binned into 5° latitude bins using a 1° grid.




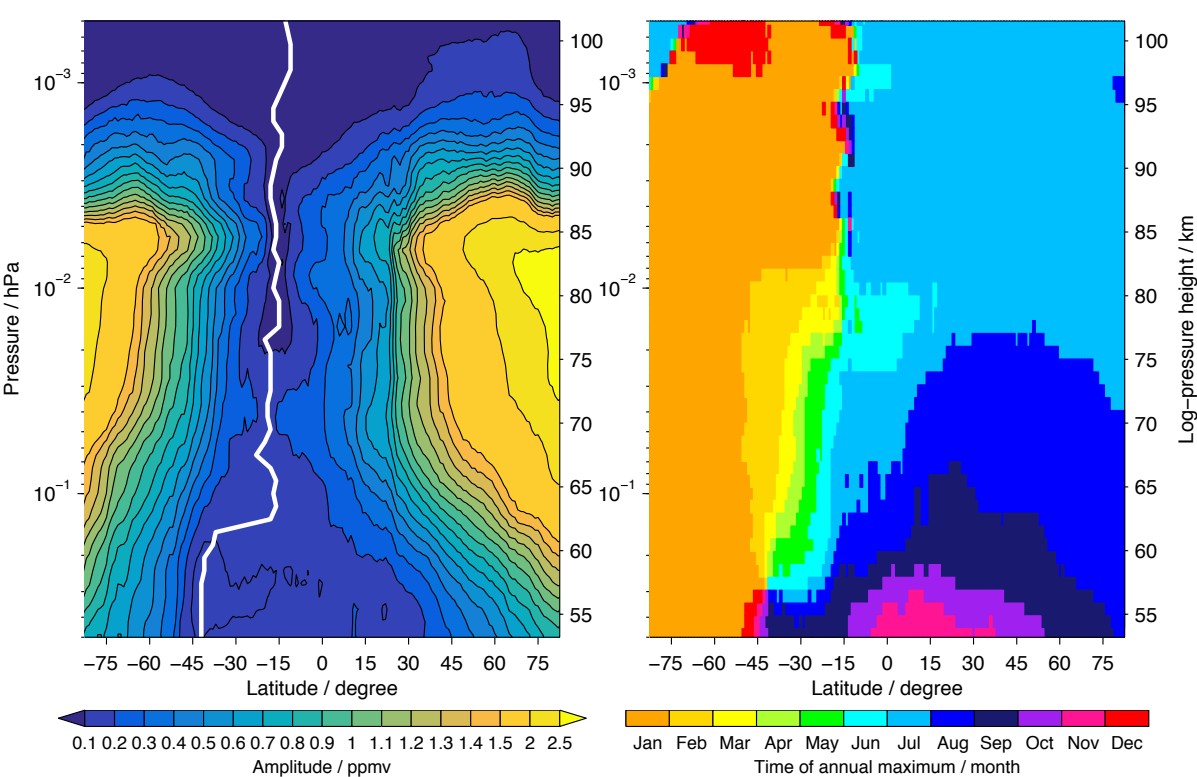

**Figure 2.** The annual variation of mesospheric water vapour based on observations of Odin/SMR (2001 – 2014). Note the different contour levels for the amplitudes as in Fig. 1. Also the latitude coverage is a bit smaller. The white line in the left panel indicates the latitude with the minimum amplitude. The data have been averaged over latitude bins of 10° due to the lower horizontal sampling of the SMR observations compared to MIPAS, but still a 1° grid is used.



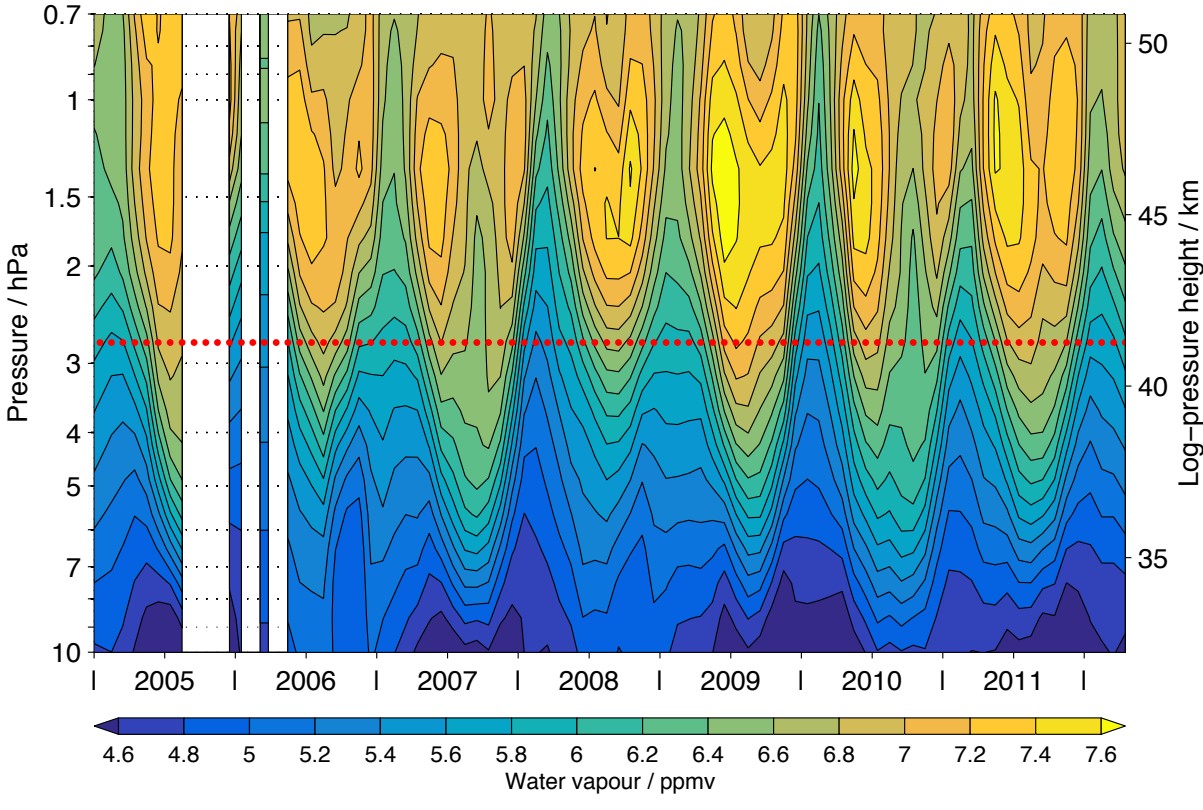

**Figure 3.** The MIPAS water vapour time series for the latitude band between 18.5°S and 13.5°S. Different to Fig. 1 only the altitude range between 10 hPa and 0.7 hPa (~32 km – 51 km) is considered here. The red dotted line shows the altitude (i.e. 2.7 hPa, ~41.5 km) where the amplitude of the annual variation in this latitude band exhibits its maximum according to Fig. 1. White areas indicate that no MIPAS data were available.





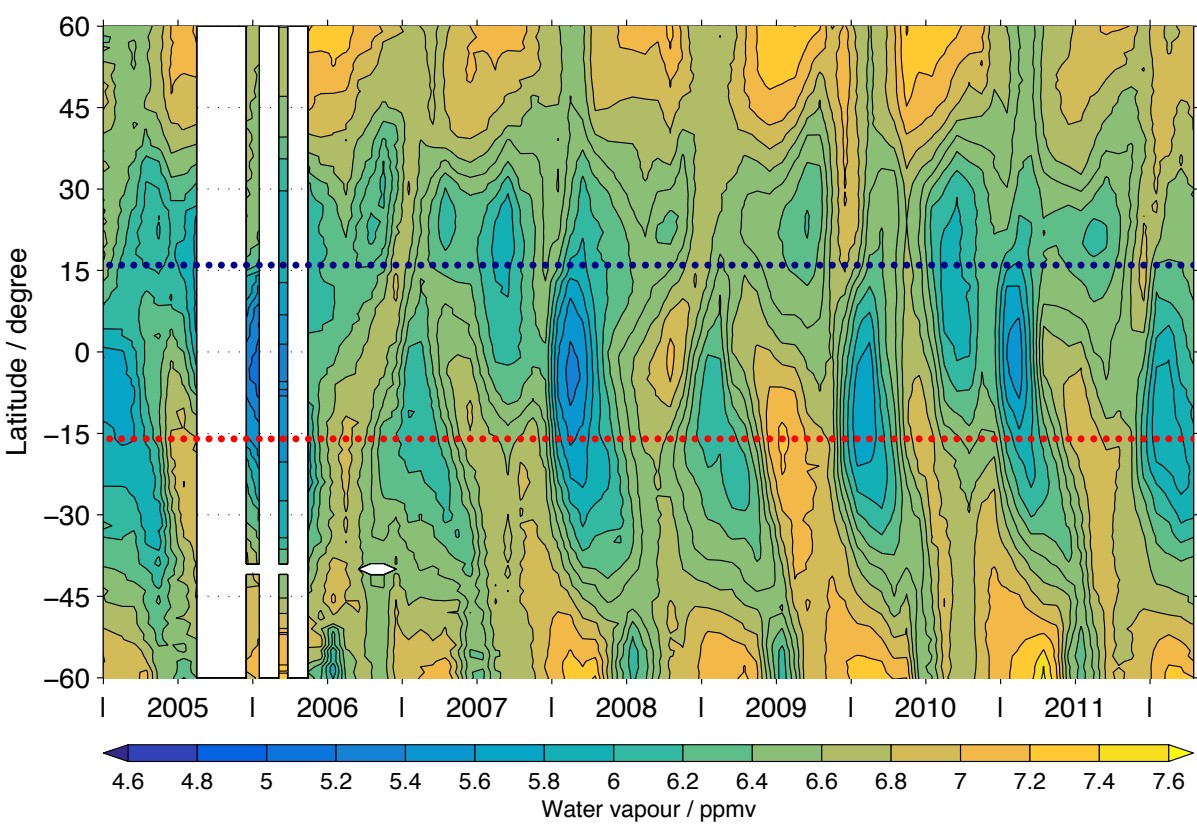

**Figure 4.** The latitude cross section from 60°S to 60°N at 2.7 hPa (∼41.5 km) as observed by MIPAS. The red dotted line indicates the latitude at which the annual variation in the southern tropics and subtropics at this pressure level maximises. For comparison a dark blue dotted line is added at the corresponding latitude in the Northern Hemisphere.





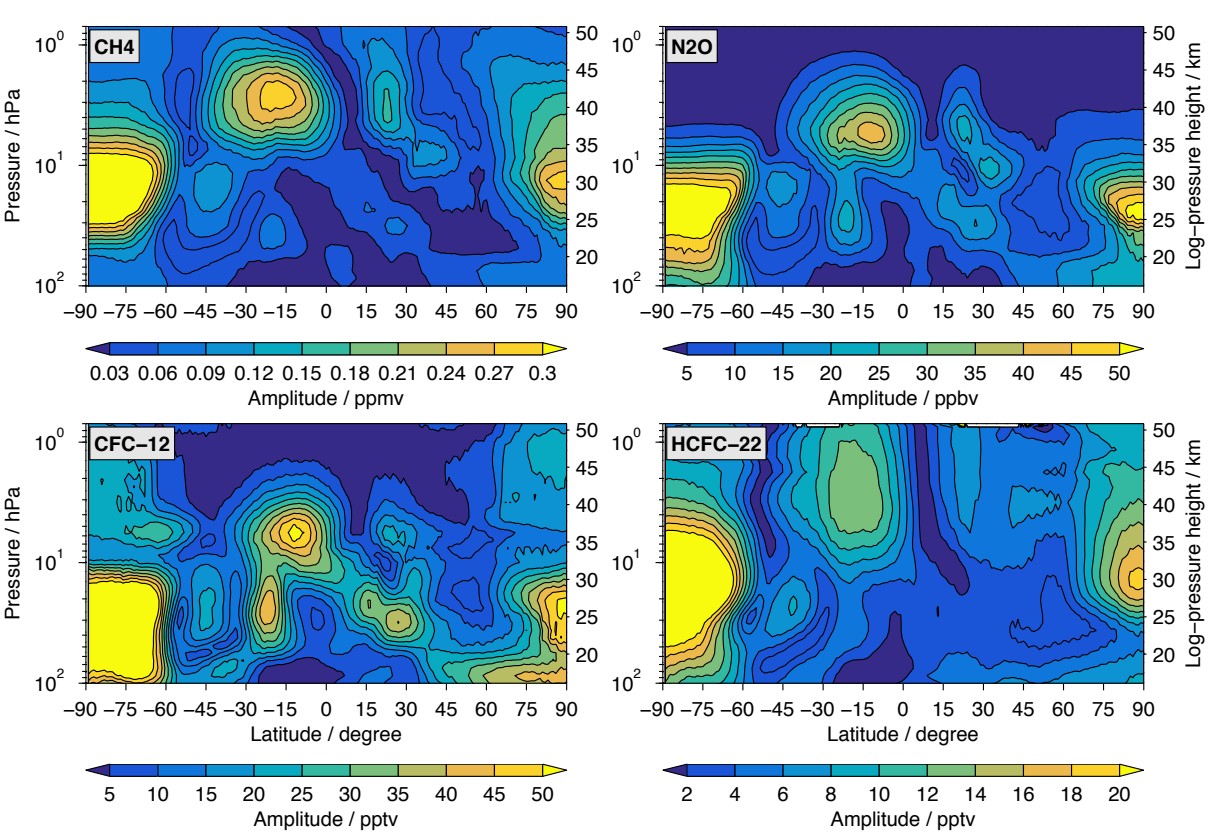

**Figure 5.** The amplitude of the annual variation in other longer-lived trace gases observed by MIPAS.





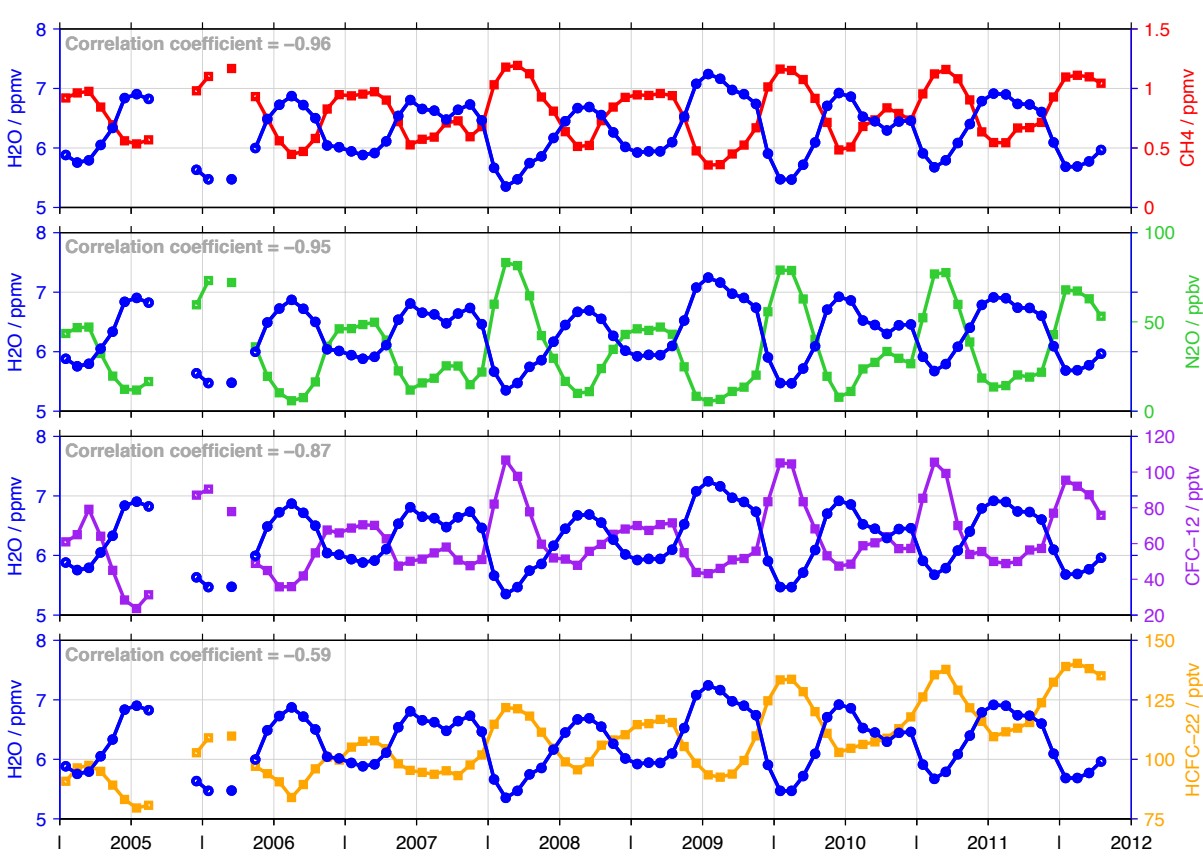

**Figure 6.** Comparison of the water vapour time series (blue, always left axis) with the time series of methane (red), nitrous oxide (green), CFC-12 (purple) and HCFC-22 (orange). All data consider MIPAS observations at 2.7 hPa for the latitude band from $18.5°$S to $13.5°$S. The correlation between the time series is indicated in the upper left corner.





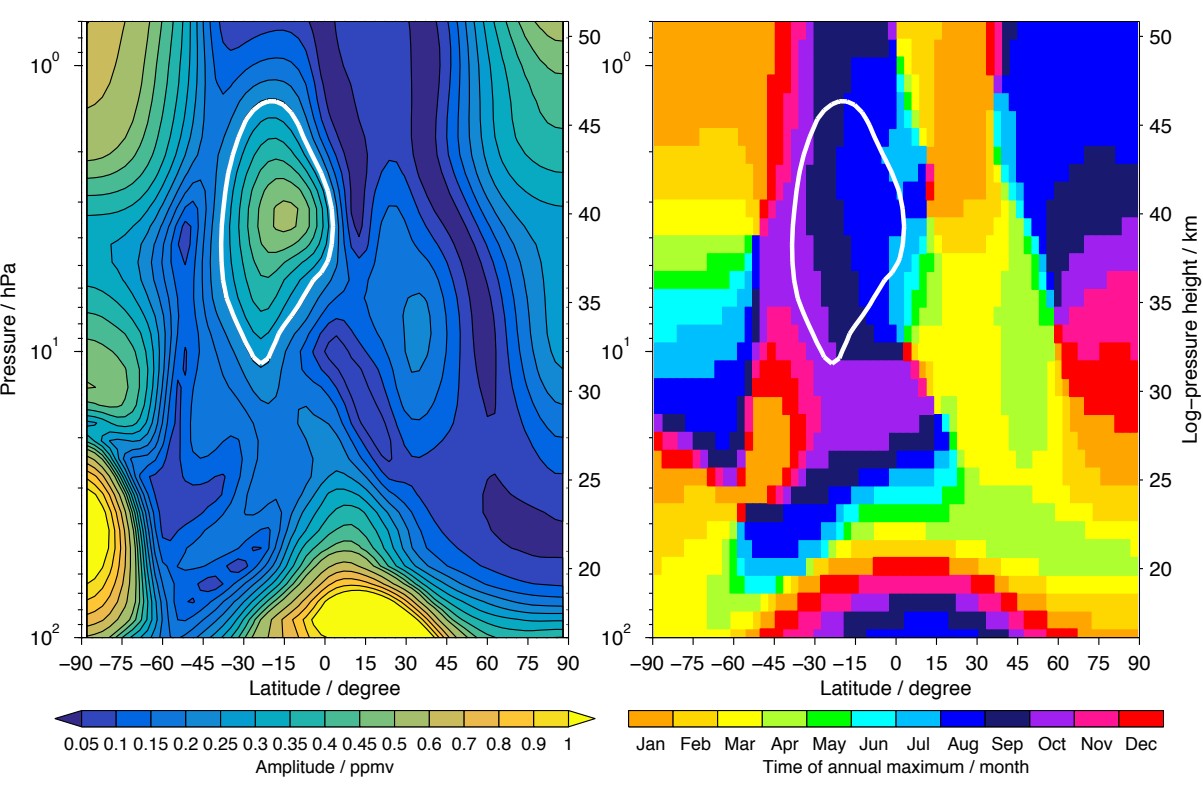

**Figure 7.** As Fig. 1 but here based on EMAC simulations for the time period from 2008 to 2012.




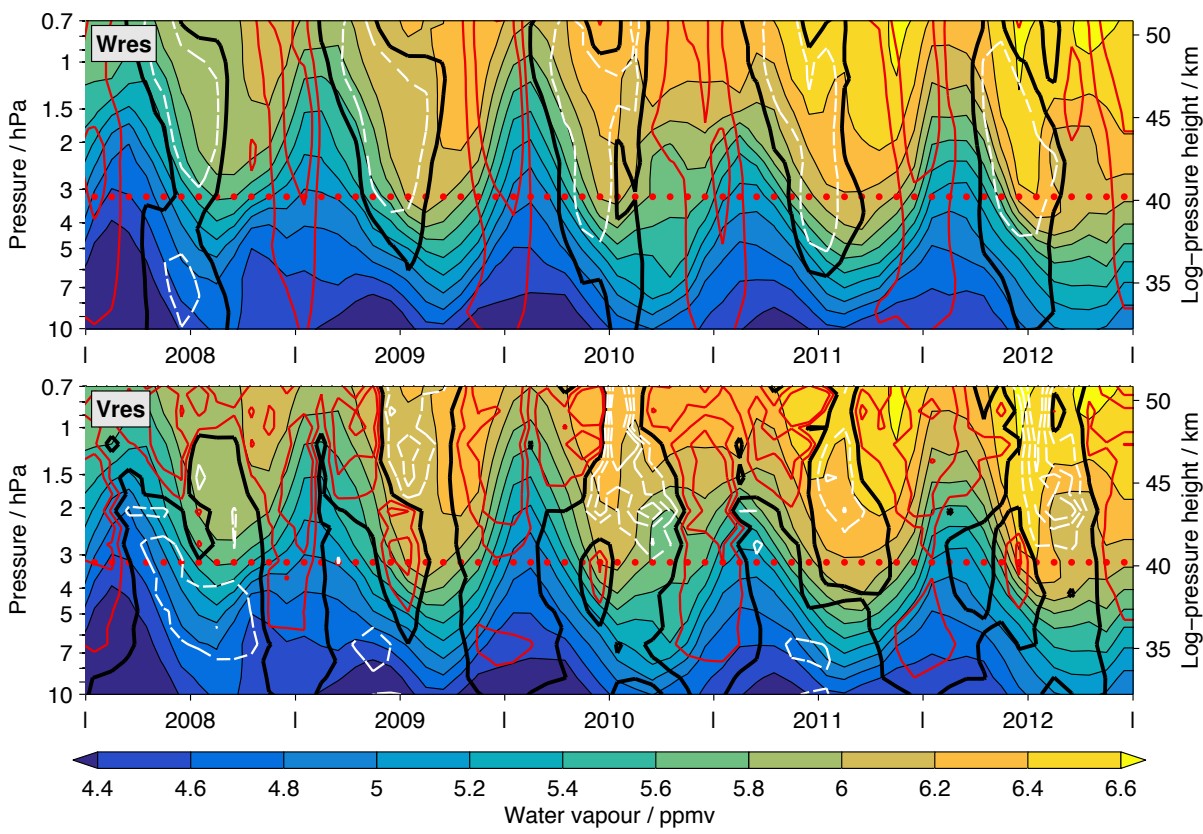

**Figure 8.** The EMAC water vapour time series at 15°S. In the upper panel the residual vertical velocity is overlaid, while in the lower panel the residual meridional velocity is considered. The black contour denotes the zero wind line. Red solid contours show positive winds (in steps of $1\,\mathrm{mm\cdot s^{-1}}$ for the vertical wind and $0.25\,\mathrm{m\cdot s^{-1}}$ for the meridional wind), while negative winds are indicated by white dashed contours (in steps of $0.5\,\mathrm{mm\cdot s^{-1}}$ for the vertical wind and $0.5\,\mathrm{m\cdot s^{-1}}$ for the meridional wind). The red dotted line marks the pressure level where the enhanced annual variation in EMAC water vapour maximises. Note the different contour levels with respect to the corresponding figure showing the MIPAS observations (Fig. 3).

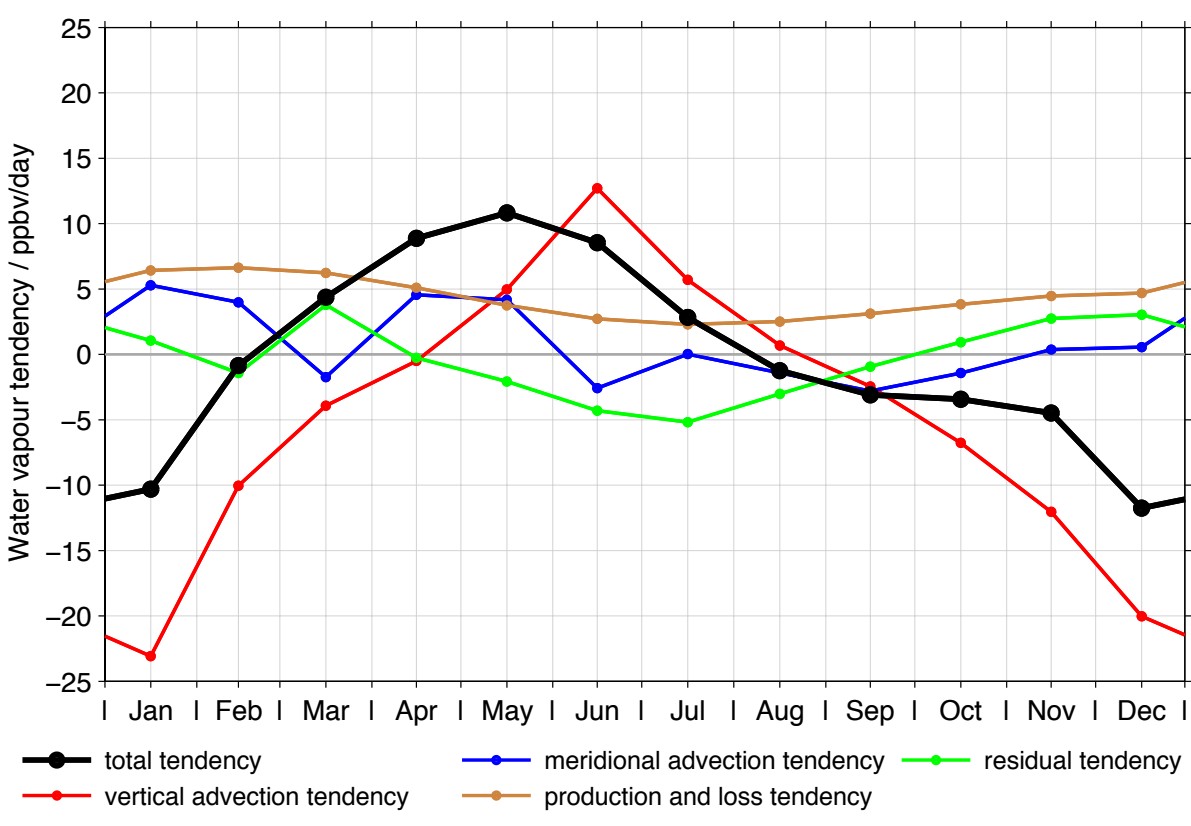

**Figure 9.** The water vapour tendencies at $15°$ S and $3.2$ hPa as derived from the EMAC simulations using climatological values based on the time period between 2008 and 2012.





**Figure 10.** An overview of the vertical structure of the different tendencies at $15°$S. The black contour indicates the zero line.





**Figure 11.** Panel (a) shows the amplitude of the annual variation of the total tendency in water vapour. As water vapour itself it shows an isolated maximum in the middle and upper stratosphere in the southern tropics and subtropics. The panels below show the annual variation amplitude for different tendencies and their combinations. It shows that only the total advection tendency in combination with the residual tendency reproduces the enhanced annual variation in its entire structure. The red dotted lines mark the peak position of the enhanced annual variation for guidance.





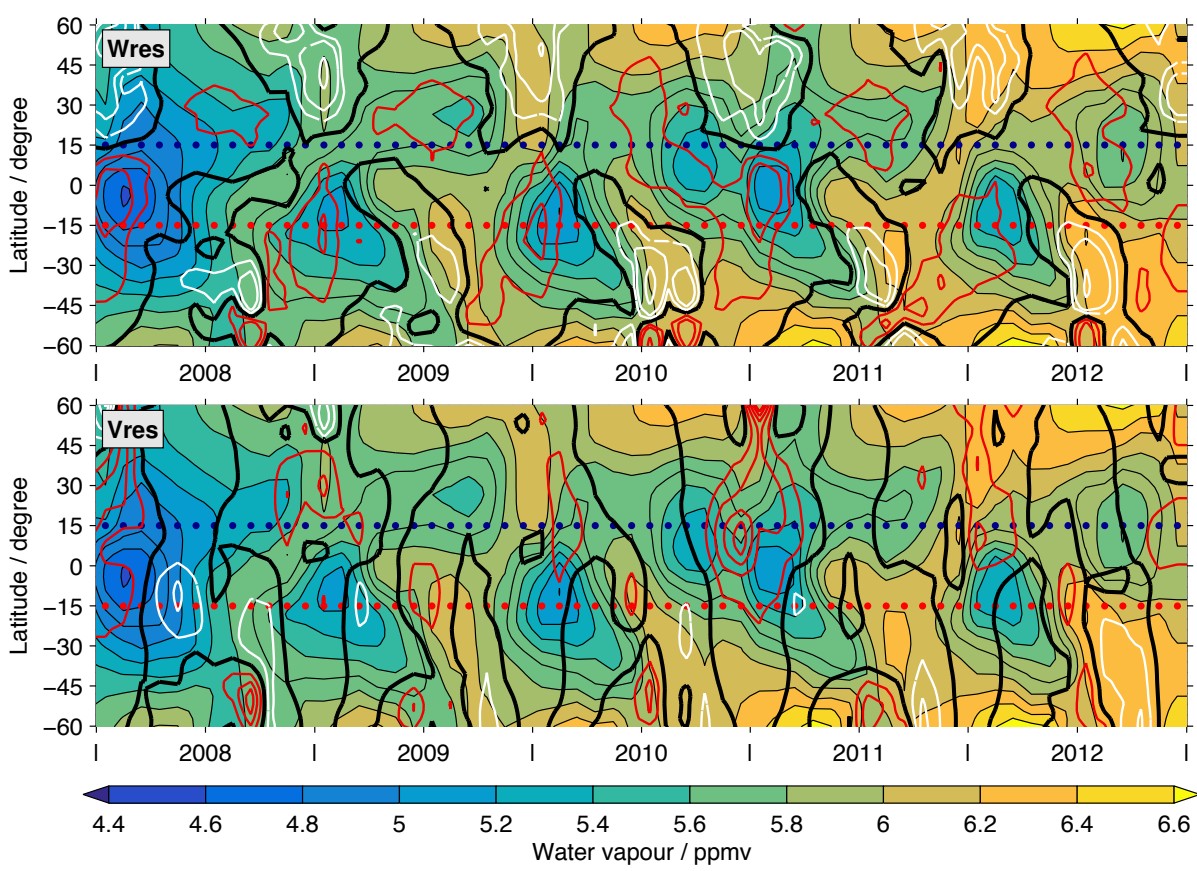

**Figure 12.** The EMAC water vapour time series at a pressure level of 3.2 hPa. As in Fig. 8 the residual vertical (upper panel) and meridional velocity (lower panel) are overlaid. The vertical winds are given at $1\,\mathrm{mm \cdot s^{-1}}$, $2\,\mathrm{mm \cdot s^{-1}}$ and $5\,\mathrm{mm \cdot s^{-1}}$ for both upwelling and downwelling. For the meridional wind contours of $0.4\,\mathrm{m \cdot s^{-1}}$ are used. The red dotted line marks indicates the latitude where the enhanced annual variation maximises. As in Fig. 4 a dark blue dotted line is added at the corresponding latitude in the Northern Hemisphere for comparison.


**Figure 13.** The latitudinal structure of the different tendencies at 3.2 hPa. The dotted lines indicate again the latitude of where the enhanced annual variation maximises (red) and the same latitude in the Northern Hemisphere (dark blue) for comparison. Note the different contours compared to Fig. 10.



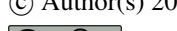

**Figure 14.** Comparison of the water vapour tendencies at 15°S/3.2 hPa and at 24°N/5.6 hPa. The results for the Northern Hemisphere have been shift by 6 months to allow a better visual comparison.





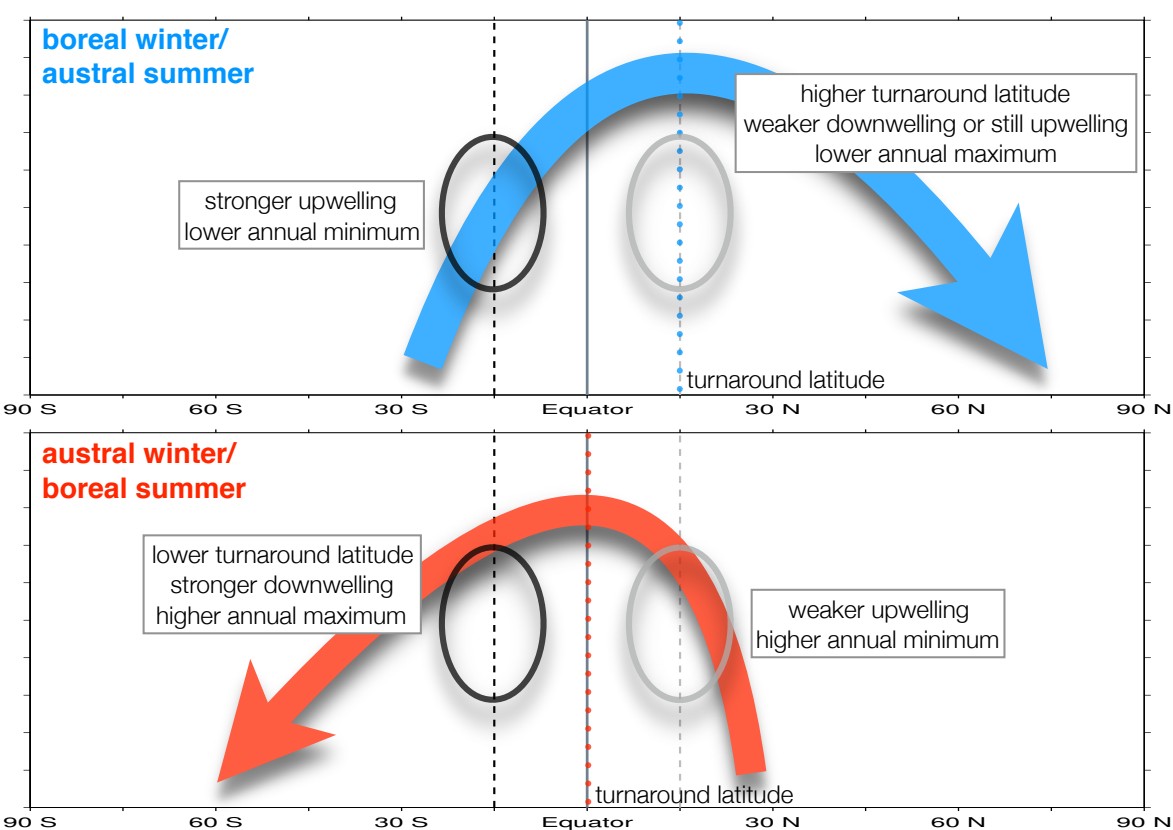

**Figure 15.** A summary sketch of the main aspects of the enhanced annual variation and the differences to its northern hemispheric counterpart with respect to the vertical advection of the Brewer-Dobson circulation. The upper panel shows the situation during boreal winter/austral summer while the lower panel focuses on austral winter/boreal summer. The region of the enhanced annual variation is depicted by the black oval. The corresponding region in the Northern Hemisphere is given in grey. The turnaround latitudes, i.e. the latitudes where the transition from upwelling to downwelling occurs, are indicated by coloured dotted lines.