# Peer review of "An "island" in the stratosphere – On the enhanced annual variation of water vapour in the middle and upper stratosphere in the southern tropics and subtropics"

_Atmospheric Chemistry and Physics, 2017_

## Referee Comment (RC1) · Anonymous Referee #1 · 23 May 2017

This paper presents some heretofore-underappreciated morphology of stratospheric tracer variability. The results are clearly explained and in agreement with the authors, may provide useful diagnostics for upper stratospheric dynamical simulations. I recommend publication subject to some minor revisions.

One general comment- the authors might consider adding some thoughts on the interannual variability. Looking at Figure 6, the AO seems to illustrate considerable variation from year to year and seems to be largest in early 2008; this appears to be captured in Figure 12. Also there is a suggestion of this in early 2011. I'm assuming some of this

would have to do with boreal winter planetary waves. Although its not straightforward since there was an SSW in January 2008, but the NH vortex was stable in 2011. Could the phase of the QBO matter? If nothing else, some discussion about this variation would provide some motivation for future work, which is always helpful.

1. Figure 12 says it's the same as Figure 8. In which case, the white contours should be dashed, not solid.

2. I found the captions for Figure 8 and 12 a bit confusing. Perhaps its only minor semantics, but I think of "time series" as single lines, as in Figure 6. The contour plots shown in Figure 8 and 12 are to me, more properly called "altitude variation as a function of time" (fig 8) and "latitudinal variation as a function of time" (fig 12). Alternatively, in Figure 4, an identical plot to figure 12 was called a "latitude cross section" which is OK. But not "time series".

---

## Referee Comment (RC2) · Anonymous Referee #2 · 6 Jun 2017

Based on both observations and simulation, this work highlights an isolated enhanced annual variation of stratospheric water vapor in the middle and upper stratosphere. The authors have clearly explained the mechanism of AO maximum primarily by the variation of deep BD circulation. This annual variation signature and its hemispheric asymmetry feature can be a useful diagnostic tool to validate models. The paper is well written and the materials sufficiently support the conclusions. I suggest publication only with a few minor revisions.

1. One concern is whether the location of the enhanced annual variation has some

shifts from year to year based on MIPAS observation and EMAC? Although the author pointed out the averaged peak location is slightly lower in EMAC simulation than MIPAS, do they have similarity of inter-annual location shifts (or amplitude)?

2. As pointed out by the first reviewer, Figure 12 should be consistent with Figure 8 where the negative winds used dashed white lines.

3. In the last schematic figure, I am confused by the grey oval showing the NH enhanced AO. I suppose it should not be symmetric with SH one (black oval) like Fig. 1 or Fig. 7. Or, the authors just intend to show the corresponding NH location to SH enhanced AO?
* * *

---

## Referee Comment (RC3) · Anonymous Referee #3 · 9 Jun 2017

Review of: An "island" in the stratosphere – On the enhanced annual variation of water vapour in the middle and upper stratosphere in the southern tropics and subtropics

by Stefan Lossow et al.

Recommendation: Minor revision.

The paper discusses a peculiar maximum in annual variability found for $H_2O$ and other tracers in the tropical upper stratosphere of both hemispheres, and the inter-hemispheric differences in this maximum (which is much stronger in the SH). I agree

with the authors that this feature may be telling us something important about transport, although, from the results presented, it is not clear what. The authors do demonstrate that mean vertical advection is the main factor that produces the annual variation (mean vertical advection has a large annual cycle, as shown for example in Figs. 9 and 10) but do not go further than this.

For example, it is not clear what processes produce this behavior (extension into the summer hemisphere of the deep branch of the BDC driven by Rossby waves in the opposite (winter) hemisphere? The details of the circulation associated with the stratopause semiannual oscillation? Something else?) Why does the EMAC model perform better when constrained by observations up to 1 hPa than when it is constrained only to 10 hPa? What might this tell us about missing or poorly represented dynamical processes in EMAC (and by extension, in other chemistry-climate models)? I would not make the elucidation of these points a requirement for publication, but the authors should try to address them insofar as possible, as this would add much value to the paper.

Specific comments (page, line number):

(2,10) "indicating an equilibrium": I don't understand why a maximum in water vapor indicates an equilibrium between sources and sinks. Wouldn't equilibrium obtain anywhere sources and sinks are balanced, regardless of whether there is a maximum in the vmr distribution? I would have thought the maximum in $H_2O$ occurs in the upper stratosphere/lower mesosphere because one runs out of $CH_4$ to oxidize and is not yet in the range of altitude where Lyman-alpha photolysis is an efficient sink.

(3,24) "large variability close to the tropopause": I would have thought the main source of variability in this region is the effect of annually-varying tropical cold-point temperature on dehydration; that is, the tape recorder signal.

(4,5) "distinct variation": Does this refer to the maximum in the SH or the NH, or both?

(4,14) "a clear inter-hemispheric difference": How does this work? What induces these differences?ÂăIn any case, I am not sure what the purpose of this figure is, other than to show that variability in the mesosphere has a different spatial distribution than in the stratosphere, which should not be a surprise, as the circulation and photochemistry are different in the two regions. If you decide going to keep this figure, then you need to explain what accounts for this pattern of variability and how that sheds light on the variability in the stratosphere. Otherwise, I would suggest you delete this material.

(5,12) "both chemical and dynamical": I don't see how chemistry is going to do this—why would chemistry (i.e., strictly speaking, reaction or photolysis rates) be different between the hemispheres?

(5,21) "importance of . . . trace gas gradients": More likely this points to the importance of photochemical sinks at higher altitude, specifically how quickly the sink increases with increasing altitude.

(6,1) "hint towards vertical advection": The conclusion "hinted at" here is not immediately obvious, as these tracers also have strong meridional gradients in the Tropics/subtropics. You later show that vertical advection is most important, but you have not done so at this point.

(6,25) "some coherent behavior": This is neither clear nor quantitative. If total hydrogen is being conserved, then neither $H_2O$ nor $CH_4$ are responding directly to transport, although the ratio of $H_2O$ to $CH_4$ will depend on the origin of the parcels transported. Or are you arguing that the transport lifetime is always shorter than the chemical lifetimes of $CH_4$ and $N_2O$ at these altitudes?

(7,24) "other tendencies have some importance": I find Fig. 9 much more useful than Fig. 8, which is too qualitative. You might consider omitting Fig. 8 altogether, assuming that the results of Fig. 9 typify what goes one throughout the SH variability maximum. Or you could make a couple more plots like Fig. 9 at other altitudes to confirm that this is the case.

(7,29) "the total tendency": Are the tendencies in Fig. 10 also composites over 2008-2012, like Fig. 9? If so, please indicate this in the figure caption and also in the text.

(8,16) "Based on the discussion so far": One thing that you have not discussed is how the semiannual variation seen in the top panel of Fig. 10 arises. It is not due to vertical advection, and it is not clear how the combination of the other term manages to produce it. Could you comment on this?

(8,21) "panel a": The panels in Fig. 11 are not labeled. Please correct.

(8,28) "eddy transport . . . residual tendency": It is an assumption, unproven as far as I can see, that the residual can be ascribed solely to (resolved or parameterized) eddy transport. This need not be the case. Numerical diffusion, for example, could play a role here.

(8,31) "yields almost the identical structure": Perhaps I have misunderstood this remark, but shouldn't this be the case, by construction?

(9,2) "relation . . . is obvious": This is not quite obvious to me. This statement refers to the circulation in the winter hemisphere at mid- and high latitudes, but here you are considering behavior in the Tropics. It is not altogether clear how the circulation in this region is driven (e.g., the relative importance of PW and GW, possible influenece of the QBO, etc.)

(9,19) "panel a": There are no panel labels in Fig. 13.

(9,29) "where the annual variation is substantial": I think you mean "where the annual variation is largest", no? That is, you compare the locations where the variation is largest in the NH and in the SH individually.

(10,34) "panel a": Again, there are no panel labels in the figure.

(11,11) "much better agreement": Please comment on why you think this is so. One possibility could be the representation of tropical dynamics, which has often been a

problem for high-top chemistry-climate models. In particular, the SAO and QBO, and their secondary circulations are not always well simulated in these models. There is not enough detail given about the model used here (EMAC), and I did not have the time to consult the literature. Perhaps some details of how tropical dynamics are handled in EMAC (in particular, the tropical GW parameterization) should be summarized briefly in Section 1.

Typos, etc.

(2, 6) "are most" are the most

(2,35) "Institut für Meteorologie"... : There is no need to put the names of these institutes in quotes.

(3,27) "provoking": I think "motivating" or "suggesting" might be better.

(4, 3) "Northern Hemisphere, however distinctively" Northern Hemisphere; however, it is distinctively...

(4,24) Figure 4: The caption of Fig. 4 reads "The latitudinal cross-section..." but does not say of what. It should be "The latitudinal cross-section of water vapor".

(5,17) "The existence in methane..." The existence of this feature in methane...

(6,14) "a tendency that" a tendency for

(7,20) "is best resembled by that" best resembles that

(8,14) "combined by" produced by the combination of

---

## Author Comment (AC1) · 14 Aug 2017

**Replies to the comments:**

We thank the reviewer for the comments. In the following, the comments are included in black while our replies are given in blue.

**General comments:**

This paper presents some heretofore-underappreciated morphology of stratospheric tracer variability. The results are clearly explained and in agreement with the authors, may provide useful diagnostics for upper stratospheric dynamical simulations. I recommend publication subject to some minor revisions.

One general comment: the authors might consider adding some thoughts on the inter-annual variability. Looking at Figure 6, the AO seems to illustrate considerable variation from year to year and seems to be largest in early 2008; this appears to be captured in Figure 12. Also there is a suggestion of this in early 2011. I'm assuming some of this would have to do with boreal winter planetary waves. Although its not straightforward since there was an SSW in January 2008, but the NH vortex was stable in 2011. Could the phase of the QBO matter? If nothing else, some discussion about this variation would provide some motivation for future work, which is always helpful.

General response: It is known that there is considerable QBO variability in the region of interest, as shown for example by Randel et al. (1998). Correspondingly, we have in our regression considered the QBO to provide a better climatological estimate of the enhanced annual variation. So far we have not investigated the inter-annual variability of the enhanced annual variation. There is a natural interest in this, since this also holds the promise to provide some information on longer-term variation, given the anticipated changes of the Brewer-Dobson circulation. However this will require multiple satellite data sets (SAGE II, HALOE, MIPAS and MLS to start with) and, thus, a careful characterisation of them to have any chance of success.

**Specific comments:**

Comment #1: Figure 12 says it's the same as Figure 8. In which case, the white contours should be dashed, not solid.

Response #1: Thanks for spotting this. It turned out to be a Matlab problem, plotting dashed lines only occasionally. It has been fixed now.

Comment #2: I found the captions for Figure 8 and 12 a bit confusing. Perhaps its only minor semantics, but I think of "time series" as single lines, as in Figure 6. The contour plots shown in Figure 8 and 12 are to me, more properly called "altitude variation as a function of time" (fig 8) and "latitudinal variation as a function of time" (fig 12). Alternatively, in Figure 4, an identical plot to figure 12 was called a "latitude cross section" which is OK. But not "time series".

Response #2: Okay, no problem. We denoted those kind of figures now as altitude-time sections or latitude-time sections. Also in the main text the terminology has been adapted.

**References:**

RANDEL, W. J., WU, F., RUSSELL, J. M., ROCHE, A., AND WATERS, J. W., "Seasonal cycles and QBO variations in stratospheric $CH_4$ and $H_2O$ observed in UARS HALOE data", *Journal of the Atmospheric Sciences*, 55, 163 – 185, doi: 10.1175/1520- 0469(1998)055<0163:SCAQVI>2.0.CO;2, 1998.

---

## Author Comment (AC2) · 14 Aug 2017

**Replies to the comments:**

We thank the reviewer for the comments. In the following, the comments are included in black while our replies are given in blue.

**General comments:**

Based on both observations and simulation, this work highlights an isolated enhanced annual variation of stratospheric water vapor in the middle and upper stratosphere. The authors have clearly explained the mechanism of AO maximum primarily by the variation of deep BD circulation. This annual variation signature and its hemispheric asymmetry feature can be a useful diagnostic tool to validate models. The paper is well written and the materials sufficiently support the conclusions. I suggest publication only with a few minor revisions.

**Specific comments:**

Comment #1: One concern is whether the location of the enhanced annual variation has some shifts from year to year based on MIPAS observation and EMAC? Although the author pointed out the averaged peak location is slightly lower in EMAC simulation than MIPAS, do they have similarity of inter-annual location shifts (or amplitude)?

Response #1: So far we have not performed any investigations beyond the climatological view. As responded to reviewer #1 there is a genuine interest to look into inter-annual variation and longer-term variability more in detail, in particular from the satellite point of view. The assumption is that with the anticipated changes of the Brewer-Dobson circulation also the enhanced annual variation will change (shift in latitude and/or altitude, size of the amplitude).

Comment #2: As pointed out by the first reviewer, Figure 12 should be consistent with Figure 8 where the negative winds used dashed white lines.

Response #2: Thanks for pointing this out. This has been fixed.

Comment #3: In the last schematic figure, I am confused by the grey oval showing the NH enhanced AO. I suppose it should not be symmetric with SH one (black oval) like Fig. 1 or Fig. 7. Or, the authors just intend to show the corresponding NH location to SH enhanced AO?

Response #3: The intention was to show the inter-hemispheric differences at 15° latitude. The caption has been rewritten to make this more obvious.

---

## Author Comment (AC3) · 14 Aug 2017

**Replies to the comments:**

We thank the reviewer for the comments. In the following, the comments are included in black while our replies are given in blue.

**General comments:**

Review of: An "island" in the stratosphere – On the enhanced annual variation of water vapour in the middle and upper stratosphere in the southern tropics and subtropics

by Stefan Lossow et al.

Recommendation: Minor revision.

The paper discusses a peculiar maximum in annual variability found for H2O and other tracers in the tropical upper stratosphere of both hemispheres, and the inter-hemispheric differences in this maximum (which is much stronger in the SH). I agree with the authors that this feature may be telling us something important about transport, although, from the results presented, it is not clear what. The authors do demonstrate that mean vertical advection is the main factor that produces the annual variation (mean vertical advection has a large annual cycle, as shown for example in Figs. 9 and 10) but do not go further than this.

For example, it is not clear what processes produce this behavior (extension into the summer hemisphere of the deep branch of the BDC driven by Rossby waves in the opposite (winter) hemisphere? The details of the circulation associated with the stratopause semiannual oscillation? Something else?) Why does the EMAC model perform better when constrained by observations up to 1 hPa than when it is constrained only to 10 hPa? What might this tell us about missing or poorly represented dynamical processes in EMAC (and by extension, in other chemistry-climate models)? I would not make the elucidation of these points a requirement for publication, but the authors should try to address them insofar as possible, as this would add much value to the paper.

General response: In this study, we focused on the reasons for the annual variation in water vapour (and other tracers), and as the reviewer states, we found the main cause to be a strong variation in upwelling in the SH subtropics.

As also stated correctly, and as sketched in Fig. 15 of the paper, this strong variation in vertical velocities in the Southern Hemisphere subtropics is due to the extension of the deep branch of the Brewer-Dobson circulation into the summer hemisphere (leading to strong upwelling in boreal winter). This extension is well known (e.g. Holton, 2012). The reviewer is right in that it would be interesting to explore the forcing of the circulation further, in particular how much of the forcing is due to local wave forcing or due to the transient response of the circulation to wave forcing in e.g. northern mid-latitudes. However, there are two reasons we decided not to go further into this:

(1) In a "nudged" model simulation (and also in reanalysis), the attribution of the circulation to forcing is sometimes not easy due to the additional nudging tendency (or increment in reanalysis, see for example Abalos et al., 2015).

(2) This goes beyond the scope of the paper and could serve as topic for an additional study in the future.

We have not included anything on the semi-annual variation, since this beyond what we intended for our manuscript. A little bit more information is found later in response #11. Response #19 comments on the problems of the EMAC simulation nudged only up to 10 hPa with regarding the enhanced annual variation.

**Specific comments (page, line number):**

Comment #1 (2,10): "indicating an equilibrium": I don't understand why a maximum in water vapor indicates an equilibrium between sources and sinks. Wouldn't equilibrium obtain anywhere sources and sinks are balanced, regardless of whether there is a maximum in the vmr distribution? I would have thought the maximum in $H_2O$ occurs in the upper stratosphere/lower mesosphere because one runs out of $CH_4$ to oxidize and is not yet in the range of altitude where Lyman-alpha photolysis is an efficient sink.

Response #1: Okay, this is poorly formulated. An equilibrium you find of course more or less everywhere. The maximum simply marks the transition from the methane oxidation dominated budget to the photodissociation dominated budget. The corresponding text in the manuscript has been adapted.

Comment #2 (3,24): "large variability close to the tropopause": I would have thought the main source of variability in this region is the effect of annually-varying tropical cold-point temperature on dehydration; that is, the tape recorder signal.

Response #2: We agree that is aspect should be mentioned as well. With "perfect" altitude resolution it is the dominant process. This information has been added to the text.

Comment #3 (4,5): "distinct variation": Does this refer to the maximum in the SH or the NH, or both?

Response #3: The entire paragraph focuses on the Northern Hemisphere. To be on the safe side this has been noted once more in the text.

Comment #4 (4,14): "a clear inter-hemispheric difference": How does this work? What induces these differences? In any case, I am not sure what the purpose of this figure is, other than to show that variability in the mesosphere has a different spatial distribution than in the stratosphere, which should not be a surprise, as the circulation and photochemistry are different in the two regions. If you decide going to keep this figure, then you need to explain what accounts for this pattern of variability and how that sheds light on the variability in the stratosphere. Otherwise, I would suggest you delete this material.

Response #4: The difference of the annual variation in the mesosphere and stratosphere has been actually the initial motivation for this study. Being used to the variability in the mesosphere the existence of such an enhanced annual variability at low latitudes in the stratosphere came a bit of a surprise, even though the dynamics and chemistry in those two atmospheric layers are arguably different. In a sense this figure pays tribute to the development of this study, but it also allows a nice overview of the annual variability in the entire middle atmosphere. It would be really a pity to remove it.

The reasons for the inter-hemispheric differences in the mesosphere lie primarily in differences in the dynamical forcing by gravity waves. This concerns on one

hand the source of gravity waves. Orography is one of the major sources at higher latitudes with obvious inter-hemispheric differences. On the other hand also the filtering of gravity waves by stratospheric winds is different in the two hemispheres. We have added references to the text on this matter.

Comment #5 (5,12): "both chemical and dynamical": I don't see how chemistry is going to do this, why would chemistry (i.e., strictly speaking, reaction or photolysis rates) be different between the hemispheres?

Response #5: This statement here considered only the situation in the Southern Hemisphere by itself, not taken into account any inter-hemispheric differences.

We would certainly expand the term "chemistry" beyond pure reaction or photolysis rate since there dependencies upon the initial amount of the constituents (i.e. $CH_4$, OH) involved or the atmospheric temperature. These dependencies may not necessarily the same in both hemispheres. In the text we used the phrase "in principal" to cover all bases, but again only focusing on the Southern Hemisphere itself.

Comment #6 (5,21): "importance of . . . trace gas gradients": More likely this points to the importance of photochemical sinks at higher altitude, specifically how quickly the sink increases with increasing altitude.

Response #6: We definitely agree with that. In the end this manifests in different gradients. The corresponding text has been adapted, naming also the chemical aspect.

Comment #7 (6,1): "hint towards vertical advection": The conclusion "hinted at" here is not immediately obvious, as these tracers also have strong meridional gradients in the Tropics/subtropics. You later show that vertical advection is most important, but you have not done so at this point.

Response #7: We have reviewed this and, yes, it could be still both vertical and meridional advection. The meridional gradients (i.e. from 5°S to 25°S) are about the same as the variation shown in Fig. 6. So, this does not help to rule out/to

prefer any of the advection terms. Thanks a lot for noting this. The text has been changed accordingly.

Comment #8 (6,25): "some coherent behavior": This is neither clear nor quantitative. If total hydrogen is being conserved, then neither H2O nor CH4 are responding directly to transport, although the ratio of H2O to CH4 will depend on the origin of the parcels transported. Or are you arguing that the transport lifetime is always shorter than the chemical lifetimes of CH4 and N2O at these altitudes?

Response #8: This was supposed to merely state that if you compare the time series of water vapour and the residual vertical velocity they have some temporal consistency, leading into a more detailed description below. We have changed the text to make this more obvious.

With regard to the lifetimes, Fig. 5.21 of Brasseur and Solomon (1986) indicates the transport lifetimes (zonal, meridional and vertical) are shorter than the photochemical lifetime. However that was not really our point here.

Comment #9 (7,24): "other tendencies have some importance": I find Fig. 9 much more useful than Fig. 8, which is too qualitative. You might consider omitting Fig. 8 altogether, assuming that the results of Fig. 9 typify what goes one throughout the SH variability maximum. Or you could make a couple more plots like Fig. 9 at other altitudes to confirm that this is the case.

Response #9: The conclusion with regard to Figs. 8 and 9 is correct. We certainly want to keep Fig. 8, since it allows some comparison with the observations. Figure 9 is clearly the answer to all our initial questions, however behind every analysis there is also a story.

We had already played with the idea to show the budget at several locations (like in Fig. 9) before the initial submission, but rejected that idea in favour of Fig. 10 and later Fig. 13. The main reason was simply that they can provide a broader overview as a focus on individual locations, even though the direct comparison of the individual budget terms is not as convenient any longer.

Comment #10 (7,29): "the total tendency": Are the tendencies in Fig. 10 also composites over 2008-2012, like Fig. 9? If so, please indicate this in the figure caption and also in the text.

Response #10: Yes they are. This information has been added to the text and the figure captions.

Comment #11 (8,16): "Based on the discussion so far": One thing that you have not discussed is how the semiannual variation seen in the top panel of Fig. 10 arises. It is not due to vertical advection, and it is not clear how the combination of the other term manages to produce it. Could you comment on this?

Response #11: In the simulations at 1 hPa the first maximum is almost entirely due to the vertical advection. The second maximum is combination of advection and the residual term. The increase from July to August is driven by the vertical advection. In later months the residual term is most important. At the same time the vertical advection yields a decrease and the decrease after September is clearly dominated by this term. However, as this is rather beyond the intended scope of our paper, we chose not to discuss this is in the paper.

Comment #12 (8,21): "panel a": The panels in Fig. 11 are not labeled. Please correct.

Response #12: The panel labels will be added by ACP later in their editing process, if you like it or not. With that a priori knowledge we already refer to them.

Comment #13 (8,28): "eddy transport . . . residual tendency": It is an assumption, unproven as far as I can see, that the residual can be ascribed solely to (resolved or parameterized) eddy transport. This need not be the case. Numerical diffusion, for example, could play a role here.

Response #13: True, ascribing the residual to eddy transport is an assumption, and it also includes any numerical diffusion. The eddy tendencies can be calculated directly in a numerical expensive way, and it has been shown that the assumption of setting the residuals to eddy transport is not perfect. However,

resolved eddy transport is the largest contribution to the residual (e.g. for carbon monoxide and ozone in the lower tropical stratosphere: Abalos et al., 2013; in the EMAC model for AoA in the lower stratosphere: Dietmüller et al., 2017). Therefore, in the context of this work (where the eddy tendencies do not play the decisive role) we argue this is a valid assumption. We mention in the text now that numerical diffusion is included as well in the residual.

Comment #14 (8,31): "yields almost the identical structure": Perhaps I have misunderstood this remark, but shouldn't this be the case, by construction?

Response #14: This remark concerns the combination of the advection and residual tendencies. As chemistry is omitted, this combination does not necessarily yield the same results as derived from the total tendency. Since, however, chemistry here only plays a minor role, the combination of the advection and residual tendencies yields almost the same results as the total tendency.

Comment #15 (9,2): "relation . . . is obvious": This is not quite obvious to me. This statement refers to the circulation in the winter hemisphere at mid- and high latitudes, but here you are considering behavior in the Tropics. It is not altogether clear how the circulation in this region is driven (e.g., the relative importance of PW and GW, possible influence of the QBO, etc.)

Response #15: We have changed the word "obvious" to "probable". The stream functions, which we did not show in the manuscript, actually do not leave any other interpretation as stated in the original version.

As stated in our general response, a complete description of the circulation forcing is beyond the intended scope of our paper.

Comment #16 (9,19): "panel a": There are no panel labels in Fig. 13.

Response #16: See response #12.

Comment #17 (9,29): "where the annual variation is substantial": I think you mean "where the annual variation is largest", no? That is, you compare the locations where the variation is largest in the NH and in the SH individually.

Response #17: No. In the Southern Hemisphere we use the location with the largest amplitude of the annual variation, but in the Northern Hemisphere we do not. Instead a lower latitude is addressed, which is closer to the latitude where the MIPAS observations show the maximum amplitude in Northern Hemisphere. Budget-wise there are only small differences to the location in the Northern Hemisphere where the annual amplitude actually maximises. Differences are primarily due to the different altitude, affecting the chemical production and loss tendency.

Comment #18 (10,34): "panel a": Again, there are no panel labels in the figure.

Response #18: See response #12.

Comment #19 (11,11): "much better agreement": Please comment on why you think this is so. One possibility could be the representation of tropical dynamics, which has often been a problem for high-top chemistry-climate models. In particular, the SAO and QBO, and their secondary circulations are not always well simulated in these models. There is not enough detail given about the model used here (EMAC), and I did not have the time to consult the literature. Perhaps some details of how tropical dynamics are handled in EMAC (in particular, the tropical GW parameterization) should be summarized briefly in Section 1.

Response #19: True, the fact that the simulation with nudging up to 1 hPa reproduces rather well the enhanced annual variation, but the free-running simulation (or nudged to lower heights) does not, clearly hints towards model deficits in the wave driving of the circulation in the model. This is why we argue in this section, that the enhanced annual feature we discuss here could have potential to identify model problems. At the same time, as discussed, the obvious reason that the fully nudged model performs quite well means that the reanalysis appears to be quite good here. As the focus here is not model

evaluation (we do not look at the free running model) we find that details of the model's wave driving are not relevant enough to be described in detail here.

**Typos, etc.:**

Comment #20 (2, 6): "are most" are the most

Response #20: Fixed.

Comment #21 (2,35): "Institut für Meteorologie". . . : There is no need to put the names of these institutes in quotes.

Response #21: The quotes have been removed.

Comment #22 (3,27): "provoking": I think "motivating" or "suggesting" might be better.

Response #22: The text has been changed from "provoking" to "motivating".

Comment #23 (4,3): "Northern Hemisphere, however distinctively" Northern Hemisphere; however, it is distinctively. . .

Response #23: We decided to go with two sentences, with slight changes to the text.

Comment #24 (4,24): Figure 4: The caption of Fig. 4 reads "The latitudinal cross-section..." but does not say of what. It should be "The latitudinal cross-section of water vapor".

Response #24: The information has been added. The caption text has been rewritten to comply with comments from another reviewer.

Comment #25 (5,17): "The existence in methane. . ." The existence of this feature in methane. . .

Response #25: The text has been changed.

Comment #26 (6,14): "a tendency that" a tendency for

Response #26: The text has been changed.

Comment #27 (7,20): "is best resembled by that" best resembles that

Response #27: We stuck with the old version, as we considered the total tendency as the reference that should be resembled and not the vertical advection tendency.

Comment #28 (8,14): "combined by" produced by the combination of

Response #20: The text has been changed.

**References:**

Brasseur, G. and Solomon, S., "Aeronomy of the middle atmosphere", D. Reidel Publishing Company, P. O. Box 17, 3300 AA Dordrecht, The Netherlands, The Second Edition, 1986.

Holton, J. R., "An introduction to dynamic meteorology", International Geophysics Series, San Diego, New York: Academic Press, 5th Edition, 2012.

Abalos, M., Randel, W. J., Kinnison, D. E. and Serrano, E., "Quantifying tracer transport in the tropical lower stratosphere using WACCM", *Atmospheric Chemistry & Physics*, 13, 10591 - 10607, https://doi.org/10.5194/acp-13-10591-2013, 2013.

Abalos, M., Legras, B., Ploeger, F. and Randel, W., "Evaluating the advective Brewer-Dobson circulation in three reanalyses for the period 1979-2012", *Journal of Geophysical Research*, 120, 2015.

Dietmüller, S., Garny, H., Ploeger, F., Jöckel, P., and Cai, D., "Effects of mixing on resolved and unresolved scales on stratospheric age of air", *Atmospheric Chemistry & Physics*, 17, 7703 - 7719, https://doi.org/10.5194/acp-17-7703-2017, 2017.